# `AttackDist`: Characterizing Zero-day Adversarial Samples by Counter Attack

## ABSTRACT

Deep Neural Networks (DNNs) have been shown vulnerable to adversarial attacks, which could produce adversarial samples that easily fool the state-of-the-art DNNs.The harmfulness of adversarial attacks calls for the defense mechanisms under fire. However, the relationship between adversarial attacks and defenses is like spear and shield.Whenever a defense method is proposed, a new attack would be followed to bypass the defense immediately.Devising a defense against new attacks (zero-day attacks) is proven to be challenging.We tackle this challenge by characterizing the intrinsic properties of adversarial samples, via measuring the norm of the perturbation after a counterattack. Our method is based on the idea that, from an optimization perspective, adversarial samples would be closer to the decision boundary; thus the perturbation to counterattack adversarial samples would be significantly smaller than normal cases. Motivated by this, we propose `AttackDist`, an attack-agnostic property to characterize adversarial samples. We first theoretically clarify under which condition `AttackDist` can provide a certified detecting performance, then show that a potential application of `AttackDist` is distinguishing zero-day adversarial examples without knowing the mechanisms of new attacks. As a proof-of-concept, we evaluate `AttackDist` on two widely used benchmarks. The evaluation results show that `AttackDist` can outperform the state-of-the-art detection measures by large margins in detecting zero-day adversarial attacks.

## 1 INTRODUCTION

Deep Neural Networks (DNNs) have flourished in recent years, and achieve outstanding performance in a lot of extremely challenging tasks, such as computer vision (He et al. (2016)), machine translation (Singh et al. (2017)), automatic speech recognition (Tüske et al. (2014)) and bioinformatics (Choi et al. (2016)). In spite of excellent performance, recent research shows that DNNs are vulnerable to adversarial samples (Dvorsky (2019)), of which the difference is unnoticeable for humans, but easily leading the DNNs to wrong predictions. This vulnerability hinders DNNs from applying in many sensitive areas, such as autonomous driving, finance, and national security.

To eliminate the impact of adversarial samples, researchers have proposed a number of techniques to help DNNs detect and prevent adversarial attacks. Existing adversarial defense techniques could be classified into two main categories: (1) adversarial robustness model retraining (Tramèr et al. (2017); Ganin et al. (2016); Shafahi et al. (2019)) and (2) statistical-based adversarial samples detection (Grosse et al. (2017); Xu et al. (2017); Meng & Chen (2017)). However, while adversarial model retraining improves defense abilities, it also leads to huge costs during retraining process, especially when the number of the parameters in current models grows larger and larger again. As for statistical-based adversarial samples detection techniques, one severe shortcoming is that all these techniques require prior knowledge about the adversarial samples, which is not realistic in most real-world cases. For example, LID (Ma et al. (2018)) and Mahalanobis (Lee et al. (2018)) need to train logic regression detectors on validation datasets. To make matters worse, adversarial attacks and defenses are just like the relationship between spear and shield. Defensive techniques that perform well against existing attacking methods will always be bypassed by new attack mechanisms, which makes defending zero-day attacks a challenging but urgent task.

To address this challenge, we propose `AttackDist`, an attack-agnostic adversarial sample detection technique via counterattack. Our method is based on insight that, from the perspective of

optimization theory, the process of searching adversarial perturbations is a non-convex optimization process. Then the adversarial perturbations generated by the attack algorithm should be close to the optimal solution $\delta^*$ (See Definition 1). Due to the property that optimal solution $\delta^*$ is close to the decision boundary (Lemma 1). Thus, if we apply the counter attack on adversarial samples, the perturbation would be significantly smaller the original samples. Figure 1 shows an example of our intuition, if we attack an adversarial sample, then the adversarial perturbation $d_2$ would be much smaller than the adversarial perturbation of attacking a normal point $d_1$. Thus by measuring the size of adversarial perturbation, we could differentiate normal points and adversarial samples.

To demonstrate the effectiveness of `AttackDist`, we first analyze the norm of adversarial perturbation for normal points and adversarial points theoretically, and give the conditions under which `AttackDist` could provide a guaranteed detecting performance (Theorem 3). In addition to theoretical analysis, we also implement `AttackDist` on two famous and widely-used benchmarks, MNIST (Deng (2012)) and Cifar-10 (Krizhevsky et al.), and compare with four state-of-the-art techniques, Vinalla (Hendrycks & Gimpel (2016)), KD (Feinman et al. (2017)), MC(Gal & Ghahramani (2016)) and Mahalanobis (Lee et al. (2018)). The experimental results show that `AttackDist` performs better than existing works in detecting zero-day adversarial attacks without requiring the prior-knowledge about the attacks.

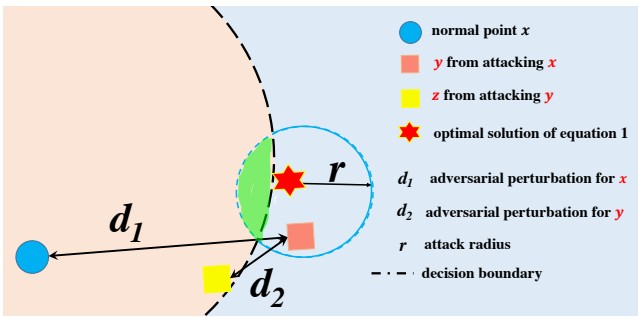

Figure 1: An example of our intuition.

In brief, we summarize our contributions as follows:

- We formally prove a general instinct property of adversarial samples (*i.e.,* adversarial samples are close to the decision boundary), which could be leveraged for detecting future advanced (less noticeable) adversarial attacks. And with more unnoticeable attacks, this property would contribute more to adversarial sample detection.

- We propose `AttackDist`, an attack-agnostic technique for detecting zero-day adversarial attacks. We theoretically prove when the adversarial perturbation satisfies the given condition, `AttackDist` could have a guaranteed performance in detecting adversarial samples.

- We implement `AttackDist` on two widely used datasets, and compare with four state-of-the-art approaches, the experiment results show `AttackDist` could achieve the state of the art performance in most cases. Especially for detecting $\ell_2$ adversarial attacks, `AttackDist` could achieve 0.99, 0.98, 0.96 AUROC score and 0.99, 0.92, 0.90 Accuracy for tree different adversarial attacks.

## 2 BACKGROUND

In this section, we first define the notations used through the paper, then give a brief review to adversarial attack and adversarial defense. Finally, we introduce our assumptions about the attackers and the defenders.

### 2.1 NOTATIONS

Let $f(\cdot) : \mathcal{X} \rightarrow \mathcal{Y}$ denote a continuous classifier, where $\mathcal{X}$ is the input space consisting of $d$-dimensional vectors, and $\mathcal{Y}$ is the output space with $K$ labels. The classifier provides prediction on a point $x$ based on $\arg\max_{r=1,2,\ldots K} f_r(x)$. We then follow () to define adversarial perturbations. Let $\Delta(\cdot)$ denote a specific attack algorithm (*e.g.,* FGSM, CW). As shown in Equation 1, given point $x$ and target classifier $f$, the adversarial perturbation $\Delta(x, f)$ provided by $\Delta(\cdot)$ is a minimal perturbation that is sufficient to change the original prediction $f(x)$ (for shorthand, we use $\Delta(x)$ to

represent $\Delta(x, f)$ throughout the paper).

$$\Delta(x, f) = min_\delta \ ||\delta||_p \qquad s.t. \quad f(x + \delta) \neq f(x) \tag{1}$$

Adversarial samples are the points that applying the adversarial perturbations on the original points (*i.e.*, $x_{adv} = x + \Delta(x)$).

**Definition 1.** *Attack Distance: We define attack distance (`AttackDist`) of a point $x$ as $\ell_p$ norm of the adversary perturbation.*

$$AttackDist(x) = ||x_{adv} - x||_p = ||\Delta(x)||_p \tag{2}$$

**Definition 2.** *Optimal Adversarial Perturbation: Given $x$ and $f$, the optimal adversarial perturbation $\delta^*(x)$ is the most optimal solution of Equation 1. In other words, $\delta^*(x)$ satisfy Equation 3.*

$$||\delta^*(x)||_p \leq ||\Delta(x)||_p \quad s.t. \quad f(x + \Delta(x)) \neq f(x) \wedge f(x + \delta^*(x)) \neq f(x) \tag{3}$$

**Definition 3.** *Optimal Adversarial Sample: Given $x$ and $f$, we define the optimal adversarial sample $x^* = x + \delta^*(x)$, that is applying the optimal adversarial perturbation $\delta^*(x)$ on normal point $x$ (Note $x^*$ is not a constant point, it is a function of $x$).*

**Definition 4.** *Decision Boundary: We define the decision boundary $B$ of classifier $f$ as the collection of points which have the same prediction on different labels. More specifically, it satisfy Equation 4.*

$$B = \{x| \ \exists i, j \ (1 \leq i, j \leq K) \wedge (i \neq j) \qquad f_{k=\{1,2,\cdots,K\}}(x) \leq f_i(x) = f_j(x)\} \tag{4}$$

Then let $D(x, f) = \min_{b \in B} \ ||x - b||_p$ (shorthand as $D(x)$) denote the minimal distance from point $x$ to the decision boundary. And we define all points on the decision boundary are adversarial samples. Because according to the definition of decision boundary $B$, any points belong to $B$ would provide more than one prediction results, which means it contains at least one prediction is contradict with the ground truth.

**Lemma 1.** *The optimal adversarial sample $x^*$ belongs to decision boundary $B$, in other words, the relationship between $\delta^*(x)$ and $D(x)$ is $||\delta^*(x)||_p = D(x)$.*

We prove lemma 1 by contradiction, assume the optimal adversarial samples $x^*$ does not belong to $B$ (*e.g.*, $x^* \notin B$), then we want to prove $x^*$ is not the most optimal adversarial samples generated from $x$ (*i.e.*, there exists $x_*$ satisfies Equation 5).

$$||x_* - x||_p < ||x^* - x||_p \qquad s.t. \quad f(x) \neq f(x_*) \tag{5}$$

*Proof.* Let $f(x) = i$, $f(x^*) = j$ and $i \neq j$. If we connect the point $x$ and $x^*$ to get a line, then there must be a point $x_*$ on the line satisfies Equation 5. We prove it by constructing the function $g(x) = f_i(x) - f_j(x)$, obviously, $g(x) > 0$ and $g(x^*) < 0$. Due to the continuous of $f$, from $x$ to $x^*$, there exists a point $g(P) = 0$ and $P \neq x^* \wedge P \neq x$. Then, we need to show that point $P$ is the point we want (*i.e.*, $P = x_*$). Obviously, $||P - x||_p < ||x_* - x||_p$ because $P$ is a middle point of straight line with $x$ and $x^*$ as endpoints, then we only need to prove $P$ would get different prediction with $x$. There are two conditions of the prediction on point $P$: (1) $f(P) = i$; (2) $f(P) \neq i$. For condition 1, due to $g(P) = 0 = f_i(P) - f_j(P)$ and $f(P) = i$, then $P$ satisfies the definition of decision boundary $B$, so $P$ would have a different prediction with $x$ (We define all points on the decision boundary are adversarial samples). For condition 2, $f(P) \neq i = f(x)$, obviously, $P$ is the adversarial sample for $x$. $\qquad \square$

Next, we introduce the definition of r-attack, to measure the the optimization capabilities of an attack algorithm. Although the definition of adversary samples is to optimize the Equation 1, but none attack algorithm could always obtain the most optimal solution $\delta^*(x)$. We define r-attack, to measure how close the adversarial samples generated by one specific attack algorithm to the optimal solution $\delta^*$.

**Definition 5.** *r-attack: we define attack algorithm $\Delta_r(\cdot)$ as an r-attack algorithm if all perturbations it produced are lying in a sphere centered on optimal adversarial perturbation with radius $r$.*

$$\Delta_r \ is \ r\text{-}attack \quad \Longleftrightarrow \quad \forall x \in \mathcal{X} \quad ||\Delta(x) - \delta^*(x)||_p \leq r \tag{6}$$

From the definition of *r-attack*, we could see more advanced attacks (more unnoticeable attack) are attacks with less $r$. The best attack could always produce $\delta^*(x)$, whose $r = 0$. As the goal of the attackers is to create less noticeable samples to evade the human-beings. Then they tend to develop the more advanced attack algorithms with smaller $r$. Later, we would show how `AttackDist` leverage this point (Theorem 3) to detect more unnoticeable attacks.

## 2.2 ADVERSARIAL ATTACKS & ADVERSARIAL DEFENSES

Many existing works have been proposed for crafting adversarial examples to fool the DNNs, we introduce a selection of such work here. The Fast Gradient Method (Goodfellow et al. (2014a)) search the adversarial samples by a small amount along the direction of gradients. The CW (Carlini & Wagner (2017)) attack, model the adversarial samples generation as a optimization problem and iteratively search the optimal solution. And the Deep Fool (Moosavi-Dezfooli et al. (2016)) attack, which is designed to estimate the distance of one sample to the decision boundary.

We then follow the definition of zero-day vulnerabilities (Ablon & Bogart (2017)) to define zero-day adversarial attacks. A zero-day adversarial attack is one attack algorithm that is unknown to those who should be interested in mitigating the attacks (*e.g.,* the adversary sample detectors).

Besides the adversarial attack techniques, a number of defense techniques also have been introduced to reduce the harms of adversarial samples. For example, KD (Feinman et al. (2017)) estimate the kernel-density of the training dataset and use the estimated kernel-density to distinguish normal samples and adversarial samples. LID, which estimate the local intrinsic dimensionality of normal, noisy and adversarial samples, and train a logic regression detector to characterize the subspace of adversarial samples. However, LID needs the prior-knowledge of adversarial attacks to train the detectors, thus can not be applied for detecting zero-day adversarial attacks.

## 2.3 THREAT MODEL

In this paper, we assume the attackers could complete access to the neural networks and could apply white-box attacks. For the detectors, they could know some attack algorithms, but when a new attack is proposed, the detectors don't know anything about the mechanism of the new proposed attacks.

## 3 APPROACH

Our aim is to gain a intrinsic properties of adversarial perturbations, and derive potential provide new directions for new advanced attacks. We begin by providing a theory analysis of the bounds of the boundary distance (`AttackDist`) of *r-attack* adversarial samples. After that, we show how `AttackDist` could be efficiently estimated through applying a counter attack. Finally, we show why `AttackDist` could differential normal samples and adversarial samples; and the condition, under which `AttackDist` could have a certificated detection performance.

## 3.1 ATTACKDIST OF ADVERSARIAL SAMPLES

Let $x$ is a normal input, we first apply algorithm $\Delta_{r1}$ to attack $x$ to generate adversary sample $y$, and apply a different algorithm $\Delta_{r2}$ to attack $y$ to generate adversary sample $z$ (*i.e.,* $y = x + \Delta_{r1}(x)$, $z = y + \Delta_{r2}(y)$). We first provide our motivation by analysing the attack distance of $x$ and $y$.

$$||y - x||_p = ||\Delta_r(x)||_p \geq ||\delta^*(x)||_p = D(x)$$
$$||\Delta_r(x)||_p \leq ||\Delta_r(x) - \delta^*(x)||_p + ||\delta^*(x,f)||_p \leq r_1 + D(x) \quad (7)$$

In the first line of Equation 7, we measure the lower bound of the adversarial perturbation. The first inequality $||\Delta_r(x)||_p \geq ||\delta^*(x)||_p$ is due to the definition of $\delta^*(x)$ (See Definition 2), and the second equality $||\delta^*(x)||_p = D(x)$ is due to Lemma 1. In the second line of Equation 7, we measure the upper bound of the adversarial perturbation. The first inequality is due to the triangle inequality (*i.e.,* $||A + B||_p \leq ||A||_p + ||B||_p$). Then because of the definition of r-attack, we have $||\Delta_{r1}(x) - \delta^*(x)||_p \leq r_1$. We then assume the random variables $D(x)$ for normal points belongs to a Gaussian distribution.

$$D(x) \sim \mathcal{N}(\mu, \sigma) \qquad \forall x \in \mathcal{X}$$

After the analysis the bound of the adversarial perturbation for normal point $x$, we then analysis the bound of the adversarial perturbation for adversarial sample $y$.

**Theorem 2.** *If $y$ is a adversary sample generated by $x$ through $r_1$-attack, and $z$ is the adversary sample generated by $y$ through $r_2$-attack then $D(y) \leq r_1$ and $||z - y||_p \leq r_1 + r_2$*

$$D(y) = min(y, B) \leq ||y - x^*||_p \leq r_1$$
$$||z - y||_p = \Delta_{r2}(y) \leq r_2 + D(y) \leq r_1 + r_2 \tag{8}$$

*Proof.* As shown in the first line in Equation 8, the distance of $y$ to decision boundary $B$ is the minimum distance of $y$ to any points in $B$. And $x^*$ is the points belongs to the decision boundary $B$ (Lemma 1), then $min(y, B) \leq ||y - x^*||_p$. And according to the definition of r-attack, $||y - x^*||_p \leq r_1$ holds. $\Delta_{r2}(y) \leq r_2 + D(y)$ in second line of Equation is because the second line of Equation 7, we just replace $x$ with $y$, and $r_1$ with $r_2$. □

**Theorem 3.** *If we have known a attack with $r_1 \leq \frac{1}{2}(\mu - 3\sigma)$, where $\mu$ and $\sigma$ are the parameters of the Gaussian Distribution for $D(x)$. Then for any advanced attacks (less noticed attack) with $r_2 \leq r_1$, we have $99.86\%$ probability that using the attack distance could correctly distinguish normal samples and adversary samples.*

*Proof.* Combining Equation 7 and 8. The lower bound for $||y - x||_p = D(x) \sim \mathcal{N}(\mu, \sigma)$, and the upper bound for $||z - y||_p$ is $r_1 + r_2$. If $r_1 \leq \frac{1}{2}(\mu - 3\sigma)$ and $r_2 \leq r_1$, then $r_1 + r_2 \leq 2r_1 \leq \mu - 3\sigma$. According to the cumulative distribution function (CDF) of Gaussian Distribution, the probability of $D(x) \leq \mu - 3\sigma$ is less than $0.14\%$. In other words, the lower bound of $||y - x||_p$ have the $99.86\%$ probability larger than the upper bound of $||z - y||_p$, which means it at least have the $99.86\%$ detection accuracy. □

### 3.2 USING ATTACKDIST TO CHARACTERIZE ADVERSARIAL EXAMPLES

We next describe how `AttackDist` can serve as property to distinguish adversarial examples without the prior-knowledge about the zero-day attacks. Our methodology only requires one known attack algorithm $\Delta_{known}$ for implementing the counter-attack. There are two main steps to calculate `AttackDist`.

1) Applying counter-attack: for the point $x$ under detection, we first attack $x$ with the known attack algorithm $\Delta_{known}$ to generate $y = x + \Delta_{known}(x)$.

2) `AttackDist` Estimation: We estimate `AttackDist` of point $x$ by measuring the norm of the adversarial perturbation $||y - x||_p$.

## 4 EVALUATION

In this section, we demonstrate the effectiveness of our method in distinguishing adversary samples generated by three attack algorithms on two widely used datasets. Our code is available at https://github.com/anonymous2021/AttackDist

### 4.1 EXPERIMENTAL SETUP

**Datasets and Models:** We evaluate our method on MNIST (Deng (2012)) and CIFAR-10 (Krizhevsky et al.) datasets. We use the standard DNN model for each dataset. For MNIST we choose LeNet-5 (LeCun et al. (2015)) architecture which reaches 98.6% accuracy on the testing set. On CIFAR-10, we train a ConNet (Carlini & Wagner (2017)) with 87.8% accuracy. The details of the model and the training setup could be found in Appendix A.

**Attack Algorithms:** We generate adversarial examples with white-box attack methods. Specifically, We consider three different attack algorithms for both $\ell_2$ and $\ell_\infty$ bounded adversarial examples. The selected attack algorithms include (see also references within):

- $\ell_2$ bounded adversarial attacks
    - BrendelBethgeAttack (BB) (Brendel et al. (2019))
    - CarliniWagnerAttack (CW) (Carlini & Wagner (2017))
    - DeepFoolAttack (DF) (Moosavi-Dezfooli et al. (2016))
- $\ell_\infty$ bounded adversarial attacks

- – ProjectedGradientDescentAttack (PGD) ( Madry et al. (2017))
- – BasicIterativeAttack (BIM) (Kurakin et al. (2016))
- – FastGradientSignAttack (FGSM) (Goodfellow et al. (2014b))

The reason we select completely different attack algorithms for $\ell_2$ and $\ell_\infty$ bounded adversarial examples is that these algorithms are designed for different $\ell_p$ norm purpose.

**Evaluation Metric:** We first point out that comparing detectors just through accuracy is not enough. For adversary samples detection, we have two classes, and the detector outputs a score for both the positive and negative classes. If the positive class is far more likely than the negative one, a detector would obtain high accuracy by always guessing the positive class, which can cause misleading results. To address this issue, besides accuracy, we also consider four different metrics. we consider the trade-off between false negatives (FN) and false positives (FP), the trade-off between precision and recall and the trade-off between true negative rate (TNR) and true positive rate (TPR), and employ Area Under the Receiver Operating Characteristic curve (**AUROC**), Area Under the Precision-Recall curve (**AUPR**), TNR at 90% true positive rate TPR (**TNR@90**) and TNR at 99% true positive rate TPR (**TNR@99**) as our evaluation metrics.

- **TNR@90** Let TP, TN, FP, and FN denote true positive, true negative, false positive and false negative, respectively. We measure TNR = TN / (FP+TN), when TPR is 90%.
- **TNR@99** We also measure TNR, when TPR is 99%.
- **AUROC** is a threshold-independent metric. The ROC curve shows the true positive rate against the false positive rate. A "perfect" detector corresponds to 100% AUROC.
- **AUPR** is also a threshold-independent metric. The PR curve plots the precision and recall against each other. A "perfect" detector has an AUPR of 100%.
- **Accuracy** We enumerate all possible thresholds $\tau$ on the test dataset and select the best accuracy for evaluation.

**Comparison Baseline:** There are many existing works could defense the adversarial attacks. However, as we discussed earlier, some of them need prior knowledge about the attacks to train the detector, as our goal is to detect zero-day adversarial attacks without the prior-knowledge, then we only consider four different approaches that require no prior knowledge of the attacks as baselines. We briefly introduce each baseline, more details about the comparison baselines could be found in related works (Hendrycks & Gimpel (2016); Feinman et al. (2017); Gal & Ghahramani (2016); Lee et al. (2018)).

- Vanilla (Hendrycks & Gimpel (2016)): Vanilla is a baseline which defines a confidence score as a maximum value of the posterior distribution. Existing works also find it could be used to detect adversary samples.
- KD (Feinman et al. (2017)): Kernel Density (KD) estimation is proposed to identify adversarial subspaces. Existing works () demonstrated the usefulness of KD-based adversary samples detection, taking advantage of the low probability density generally associated with adversarial subspaces.
- MC(Gal & Ghahramani (2016)): MC Drop represeent the model uncertainty for a specific input activate the dropout layer in the testing phase.
- Mahalanobis (Lee et al. (2018)): Mahalanobis using Mahalanobis distance on the features (the output in the hidden layer) learned by the target DNNs to distinguish adversarial samples, it is an approach based on uncertainty measurement.

For the baseline KD, it needs to tune the hyperparameters for computing. We follow (Ma et al. (2018)) to set the optimal bandwidths chosen for MNIST, CIFAR-10 as 3.79 and 0.26, respectively. As for MC, we activate the dropout layer and run 300 times. For Mahalanobis, it need selects the features in the hidden layer to a Gaussian model. For MNIST, we select features before the last fully connected layer, and for CIFAR-10, we select the last two layers.

**Experiment Process:**

Since our approach is based on counterattack, we need to use a known attack algorithm during the implementation, which is easy to meet because there are a variety of open-source attack algorithms.

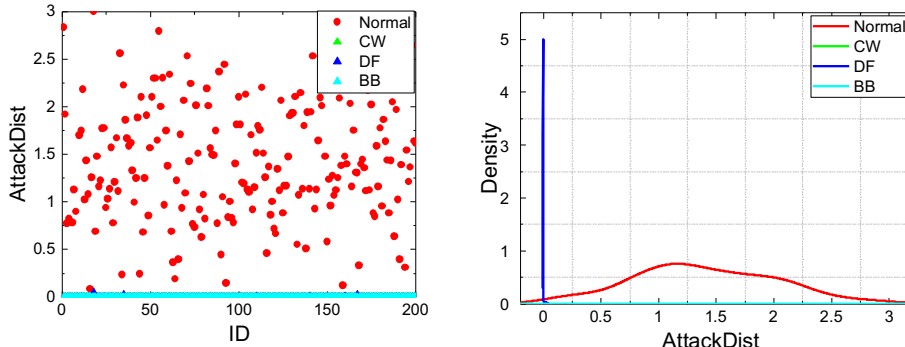

Figure 2: The Distribution and Probability density function of `AttackDist` of two hundreds random selected data

Table 1: The Experiment Results of $\ell_2$ norm Attacks On CIFAR-10 Dataset

| Attack | Baselines | TNR@99 | | | | TNR@90 | | | | AUROC | | | | AUPR | | | | Acc | | | |
|---|---|---|---|---|---|---|---|---|---|---|---|---|---|---|---|---|---|---|---|---|---|
| | | BB | CW | DF | Ave | BB | CW | DF | Ave | BB | CW | DF | Ave | BB | CW | DF | Ave | BB | CW | DF | Ave |
| BB | Vinalla | 0.00 | 0.00 | 0.00 | 0.00 | 0.01 | 0.01 | 0.01 | 0.01 | 0.95 | 0.95 | 0.95 | 0.95 | 0.31 | 0.31 | 0.31 | 0.31 | 0.50 | 0.50 | 0.50 | 0.50 |
| | MC | 0.00 | 0.00 | 0.00 | 0.00 | 0.01 | 0.01 | 0.01 | 0.01 | 0.97 | 0.97 | 0.97 | 0.97 | 0.32 | 0.32 | 0.32 | 0.32 | 0.50 | 0.50 | 0.50 | 0.50 |
| | KD | 0.00 | 0.00 | 0.00 | 0.00 | 0.01 | 0.01 | 0.01 | 0.01 | 0.72 | 0.72 | 0.72 | 0.72 | 0.37 | 0.37 | 0.37 | 0.37 | 0.50 | 0.50 | 0.50 | 0.50 |
| | Ma | 0.05 | 0.05 | 0.05 | 0.05 | 0.16 | 0.15 | 0.16 | 0.16 | 0.51 | 0.51 | 0.51 | 0.51 | 0.49 | 0.49 | 0.49 | 0.49 | 0.53 | 0.53 | 0.53 | 0.53 |
| | ours | 0.64 | 0.74 | 0.80 | 0.72 | 1.00 | 1.00 | 1.00 | 1.00 | 0.99 | 0.99 | 0.99 | 0.99 | 0.99 | 1.00 | 1.00 | 0.99 | 0.98 | 0.99 | 0.99 | 0.99 |
| CW | Vinalla | 0.00 | 0.00 | 0.00 | 0.00 | 0.02 | 0.02 | 0.02 | 0.02 | 0.94 | 0.94 | 0.94 | 0.94 | 0.32 | 0.32 | 0.32 | 0.32 | 0.50 | 0.50 | 0.50 | 0.50 |
| | MC | 0.00 | 0.00 | 0.00 | 0.00 | 0.01 | 0.01 | 0.01 | 0.01 | 0.96 | 0.96 | 0.96 | 0.96 | 0.32 | 0.32 | 0.32 | 0.32 | 0.50 | 0.50 | 0.50 | 0.50 |
| | KD | 0.00 | 0.00 | 0.00 | 0.00 | 0.02 | 0.02 | 0.02 | 0.02 | 0.70 | 0.70 | 0.70 | 0.70 | 0.37 | 0.37 | 0.37 | 0.37 | 0.50 | 0.50 | 0.50 | 0.50 |
| | Ma | 0.04 | 0.04 | 0.04 | 0.04 | 0.16 | 0.16 | 0.16 | 0.16 | 0.51 | 0.51 | 0.51 | 0.51 | 0.50 | 0.50 | 0.50 | 0.50 | 0.53 | 0.53 | 0.54 | 0.53 |
| | ours | 0.63 | 0.75 | 0.74 | 0.70 | 0.93 | 0.94 | 0.94 | 0.94 | 0.98 | 0.98 | 0.98 | 0.98 | 0.98 | 0.98 | 0.98 | 0.98 | 0.92 | 0.92 | 0.92 | 0.92 |
| DF | Vinalla | 0.00 | 0.00 | 0.00 | 0.00 | 0.02 | 0.02 | 0.02 | 0.02 | 0.93 | 0.93 | 0.93 | 0.93 | 0.32 | 0.32 | 0.32 | 0.32 | 0.50 | 0.50 | 0.50 | 0.50 |
| | MC | 0.00 | 0.00 | 0.00 | 0.00 | 0.01 | 0.01 | 0.01 | 0.01 | 0.94 | 0.94 | 0.94 | 0.94 | 0.32 | 0.32 | 0.32 | 0.32 | 0.50 | 0.50 | 0.50 | 0.50 |
| | KD | 0.00 | 0.00 | 0.00 | 0.00 | 0.03 | 0.03 | 0.03 | 0.03 | 0.69 | 0.69 | 0.69 | 0.69 | 0.38 | 0.38 | 0.38 | 0.38 | 0.50 | 0.50 | 0.50 | 0.50 |
| | Ma | 0.05 | 0.05 | 0.05 | 0.05 | 0.19 | 0.17 | 0.19 | 0.18 | 0.52 | 0.52 | 0.52 | 0.52 | 0.49 | 0.50 | 0.50 | 0.50 | 0.54 | 0.54 | 0.55 | 0.54 |
| | ours | 0.34 | 0.39 | 0.42 | 0.39 | 0.89 | 0.89 | 0.89 | 0.89 | 0.95 | 0.96 | 0.96 | 0.96 | 0.95 | 0.95 | 0.96 | 0.95 | 0.89 | 0.90 | 0.90 | 0.90 |
| Mix | Vinalla | 0.00 | 0.00 | 0.00 | 0.00 | 0.02 | 0.02 | 0.02 | 0.02 | 0.94 | 0.94 | 0.94 | 0.94 | 0.55 | 0.55 | 0.55 | 0.55 | 0.75 | 0.75 | 0.75 | 0.75 |
| | MC | 0.00 | 0.00 | 0.00 | 0.00 | 0.01 | 0.01 | 0.01 | 0.01 | 0.96 | 0.96 | 0.96 | 0.96 | 0.55 | 0.55 | 0.55 | 0.55 | 0.75 | 0.75 | 0.75 | 0.75 |
| | KD | 0.00 | 0.00 | 0.00 | 0.00 | 0.02 | 0.02 | 0.02 | 0.02 | 0.70 | 0.70 | 0.70 | 0.70 | 0.63 | 0.63 | 0.63 | 0.63 | 0.75 | 0.75 | 0.75 | 0.75 |
| | Ma | 0.04 | 0.05 | 0.04 | 0.04 | 0.16 | 0.16 | 0.16 | 0.16 | 0.51 | 0.51 | 0.51 | 0.51 | 0.75 | 0.75 | 0.75 | 0.75 | 0.76 | 0.76 | 0.76 | 0.76 |
| | ours | 0.54 | 0.51 | 0.58 | 0.55 | 0.93 | 0.94 | 0.94 | 0.94 | 0.97 | 0.98 | 0.98 | 0.98 | 0.99 | 0.99 | 0.99 | 0.99 | 0.93 | 0.94 | 0.94 | 0.93 |

In our experiments, we literally treat one attack algorithms as known attack algorithm, and the other two algorithms as the zero-day attacks to generate adversary samples, then evaluate whether `AttackDist` could detect the generated zero-day adversary samples. For example, we use PGDM as the known attack and treat it as our approach's input, then use the rest attack algorithms to generate zero-day adversarial samples for evaluation. The detail implementation of our attack and attack success rates can be find in Appendix B.

## 4.2 BoundaryDist Characteristics of Adversarial samples

We provide empirical results showing the `AttackDist` characteristics of adversarial samples crafted by the mentioned attacks. We use CW attack algorithm to counter attack the adversarial samples generated by CW, DF, BB and the normal samples, and measure the `AttackDist` to show how `AttackDist` could distinguish normal samples and adversarial samples. The left subfigure in Figure 2 shows the `AttackDist` of 200 randomly selected normal, and adversarial examples from the MNIST dataset. Left figure shows the $\ell_2$ norm attack and right figure shows the $\ell_{inf}$ norm attack. Red circle points represent the normal points, while different color square points represents the different adversarial samples. We observe that `AttackDist` scores of adversarial examples are significantly smaller than those of normal examples, especially for the $\ell_{inf}$ norm attacks. This supports our expectation that the perturbation to counterattack adversarial samples would be significantly smaller than normal cases. The right subfigure in Figure 2 shows the probability density function (PDF) of normal, and adversarial examples. Clearly, the distribution of normal samples and adversarial smaples are totally different. The different PDFs suggest that by selecting a property threshold, `AttackDist` could correctly detect the adversarial samples.

## 4.3 Experimental Results

Due to the limit of space, we only present the results for CIFAR-10, the results about MNIST could be found in Appendix C.

### 4.3.1 $\ell_2$ ATTACK

Table 1 shows our experimental results to detect $\ell_2$ norm adversarial attacks on CIFAR-10 dataset. For almost all cases, our approach outperforms the baselines for great margins, especially for NTR@99, when the requirement of TNR is 0.99, the performances of baselines are almost zero, which means the existing works fail to detect the new adversarial attacks without the prior-knowledge. However, `AttackDist` could still have a a 0.54, 051, 0.58 NTR for mixed adversarial attacks. As for the metric NTR@90, which is a slightly loose requirement than NTR99. At this scenario, the performance of the baseline is no-longer zero, however, they still have a poor performance, while `AttackDist` almost have a perfect performance with 0.93, 0.94, 0.94 for mixed attacks. Another interesting finding is that `AttackDist` almost keep the same performance with different attack algorithms we choose to implementing the counter-attack. This means `AttackDist` is not sensitive with the adversarial attack algorithms for counter-attacking.

### 4.3.2 $\ell_\infty$ ATTACK

Table 2: The Experiment Results of $\ell_\infty$ norm Attacks On CIFAR-10 Dataset

| Attack | Baselines | TNR@99 | | | | TNR@90 | | | | AUROC | | | | AUPR | | | | Acc | | | |
|---|---|---|---|---|---|---|---|---|---|---|---|---|---|---|---|---|---|---|---|---|---|
| | | BB | CW | DF | Ave | BB | CW | DF | Ave | BB | CW | DF | Ave | BB | CW | DF | Ave | BB | CW | DF | Ave |
| BB | Vinalla | 0.02 | 0.02 | 0.02 | 0.02 | 0.19 | 0.19 | 0.19 | 0.19 | 0.57 | 0.57 | 0.57 | 0.57 | 0.53 | 0.53 | 0.53 | 0.53 | 0.57 | 0.57 | 0.57 | 0.57 |
| | MC | 0.02 | 0.02 | 0.02 | 0.02 | 0.18 | 0.18 | 0.18 | 0.18 | 0.58 | 0.58 | 0.58 | 0.58 | 0.55 | 0.55 | 0.55 | 0.55 | 0.57 | 0.57 | 0.57 | 0.57 |
| | KD | 0.00 | 0.00 | 0.00 | 0.00 | 0.08 | 0.08 | 0.08 | 0.08 | 0.54 | 0.54 | 0.54 | 0.54 | 0.47 | 0.47 | 0.47 | 0.47 | 0.50 | 0.50 | 0.50 | 0.50 |
| | Ma | 0.02 | 0.01 | 0.01 | 0.01 | 0.10 | 0.10 | 0.09 | 0.10 | 0.51 | 0.50 | 0.50 | 0.50 | 0.51 | 0.51 | 0.50 | 0.51 | 0.53 | 0.53 | 0.52 | 0.52 |
| | ours | 0.06 | 0.05 | 0.19 | 0.10 | 0.18 | 0.17 | 0.18 | 0.18 | 0.54 | 0.55 | 0.54 | 0.54 | 0.45 | 0.45 | 0.45 | 0.45 | 0.55 | 0.55 | 0.55 | 0.55 |
| CW | Vinalla | 0.06 | 0.06 | 0.06 | 0.06 | 0.29 | 0.29 | 0.29 | 0.29 | 0.67 | 0.67 | 0.67 | 0.67 | 0.63 | 0.63 | 0.63 | 0.63 | 0.64 | 0.64 | 0.64 | 0.64 |
| | MC | 0.04 | 0.04 | 0.04 | 0.04 | 0.25 | 0.25 | 0.25 | 0.25 | 0.65 | 0.65 | 0.65 | 0.65 | 0.60 | 0.60 | 0.60 | 0.60 | 0.63 | 0.63 | 0.63 | 0.63 |
| | KD | 0.00 | 0.00 | 0.00 | 0.00 | 0.07 | 0.07 | 0.07 | 0.07 | 0.59 | 0.59 | 0.59 | 0.59 | 0.43 | 0.43 | 0.43 | 0.43 | 0.50 | 0.50 | 0.50 | 0.50 |
| | Ma | 0.01 | 0.01 | 0.01 | 0.01 | 0.08 | 0.08 | 0.06 | 0.07 | 0.50 | 0.50 | 0.50 | 0.50 | 0.53 | 0.54 | 0.52 | 0.53 | 0.54 | 0.55 | 0.54 | 0.55 |
| | ours | 0.06 | 0.04 | 0.19 | 0.10 | 0.14 | 0.13 | 0.19 | 0.15 | 0.62 | 0.62 | 0.61 | 0.62 | 0.41 | 0.41 | 0.41 | 0.41 | 0.53 | 0.53 | 0.52 | 0.53 |
| DF | Vinalla | 0.00 | 0.00 | 0.00 | 0.00 | 0.03 | 0.03 | 0.03 | 0.03 | 0.69 | 0.69 | 0.69 | 0.69 | 0.38 | 0.38 | 0.38 | 0.38 | 0.50 | 0.50 | 0.50 | 0.50 |
| | MC | 0.00 | 0.00 | 0.00 | 0.00 | 0.04 | 0.04 | 0.04 | 0.04 | 0.68 | 0.68 | 0.68 | 0.68 | 0.39 | 0.39 | 0.39 | 0.39 | 0.50 | 0.50 | 0.50 | 0.50 |
| | KD | 0.00 | 0.00 | 0.00 | 0.00 | 0.07 | 0.07 | 0.07 | 0.07 | 0.55 | 0.55 | 0.55 | 0.55 | 0.46 | 0.46 | 0.46 | 0.46 | 0.50 | 0.50 | 0.50 | 0.50 |
| | Ma | 0.02 | 0.01 | 0.03 | 0.02 | 0.12 | 0.11 | 0.16 | 0.13 | 0.50 | 0.52 | 0.51 | 0.51 | 0.48 | 0.47 | 0.50 | 0.48 | 0.52 | 0.51 | 0.54 | 0.52 |
| | ours | 0.06 | 0.06 | 0.19 | 0.10 | 0.33 | 0.33 | 0.35 | 0.34 | 0.70 | 0.70 | 0.70 | 0.70 | 0.64 | 0.64 | 0.64 | 0.64 | 0.65 | 0.66 | 0.65 | 0.65 |
| Mix | Vinalla | 0.01 | 0.01 | 0.01 | 0.01 | 0.10 | 0.10 | 0.10 | 0.10 | 0.52 | 0.52 | 0.52 | 0.52 | 0.76 | 0.76 | 0.76 | 0.76 | 0.75 | 0.75 | 0.75 | 0.75 |
| | MC | 0.01 | 0.01 | 0.01 | 0.01 | 0.09 | 0.09 | 0.09 | 0.09 | 0.52 | 0.52 | 0.52 | 0.52 | 0.76 | 0.76 | 0.76 | 0.76 | 0.75 | 0.75 | 0.75 | 0.75 |
| | KD | 0.00 | 0.00 | 0.00 | 0.00 | 0.07 | 0.07 | 0.07 | 0.07 | 0.56 | 0.56 | 0.56 | 0.56 | 0.71 | 0.71 | 0.71 | 0.71 | 0.75 | 0.75 | 0.75 | 0.75 |
| | Ma | 0.01 | 0.01 | 0.01 | 0.01 | 0.10 | 0.10 | 0.10 | 0.10 | 0.50 | 0.50 | 0.50 | 0.50 | 0.76 | 0.75 | 0.76 | 0.75 | 0.75 | 0.75 | 0.75 | 0.75 |
| | ours | 0.06 | 0.04 | 0.19 | 0.10 | 0.16 | 0.17 | 0.20 | 0.18 | 0.51 | 0.51 | 0.52 | 0.51 | 0.75 | 0.75 | 0.75 | 0.75 | 0.76 | 0.76 | 0.73 | 0.75 |

Table 2 shows our experimental results to detect $\ell_\infty$ norm adversarial attacks on CIFAR-10 dataset. The performance of detecting $\ell_\infty$ norm adversarial attacks is much worse than $\ell_2$ attacks. However, `AttackDist` still achieve a competitive performance, one possible reason that `AttackDist` can not have a good performance as $\ell_2$ attacks is the condition in Theorem 3 is no longer hold for $\ell_\infty$ attacks. The existing works (Carlini & Wagner (2017)) studied the size of adversarial perturbation for $\ell_2$ and $\ell_\infty$ attacks. On CIFAR-10 dataset, $\ell_\infty = 0.013$ is enough to achieve the average 100% success attack rate, while $\ell_2$ needs to be larger than 0.33. However, consider the different maximum distance on $\ell_2$ and $\ell_\infty$ norm (*i.e.,* the maximum $\ell_2$ norm for CIFAR-10 is $32 \times 32 \times 3 = 3072$ while the maximum $\ell_\infty$ is 1). Then the relative $r$ for $\ell_2$ norm attacks would be smaller, which means the $\ell_2$ could produce more unnoticeable adversarial samples.

## 5 DISCUSSIONS AND CONCLUSIONS

In this paper, we proposed `AttackDist` to address the challenge of detecting zero-day adversarial attacks. From the perspective of optimization theory, we try to understand the general intrinsic properties of adversarial samples rather than statistically analysis the hidden feature of existing adversarial samples. Trough counter attack the normal samples and the adversarial samples, we analysis the norm of the adversarial perturbation of normal samples and adversarial samples theoretically, and give the condition under which `AttackDist` would have a guaranteed performance for any advanced attacks. In particular, `AttackDist` performs better than the existing works in detecting zero-day adversarial samples.

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

## A  MODEL ARCHITECTURE AND TRAINING SETUP

Table 3: The Model Architecture we used in our experiments

| Layer Type | MNIST | CIFAR-10 |
|---|---|---|
| Convolution + ReLU | $5 \times 5 \times 6$ | $3 \times 3 \times 64$ |
| Convolution + ReLU | | $3 \times 3 \times 64$ |
| Max Pooling | $2 \times 2$ | $2 \times 2$ |
| Convolution + ReLU | $5 \times 5 \times 16$ | $3 \times 3 \times 128$ |
| Convolution + ReLU | | $3 \times 3 \times 128$ |
| Max Pooling | $2 \times 2$ | $2 \times 2$ |
| Fully Connected + ReLU | 128 | 128 |
| Fully Connected + ReLU | 80 | 128 |
| softmax | 10 | 10 |

The model architecture we used is listed in Table 3, for MNIST and CIFAR-10, the training dataset is 50,000 and we randomly select 1,000 from the testing dataset for evaluation. We set the learning rate as 0.01, with the momentum is 0.9.

# B   ADVERSARIAL SAMPLES GENERATION

We implement the listed attack algorithms through the python library `foolbox` (Rauber et al. (2020)), which is a popular library for evaluating the robustness of DNNs. For MNIST dataset, we set hyper-parameter $\epsilon$ as 3 and 0.25 for $\ell_2$ and $\ell_{\inf}$; for CIFAR-10 dataset, we set $\epsilon$ as 0.35 and 0.015 for $\ell_2$ and $\ell_{\inf}$. The attack success rates for each attack algorithms are listed in Table 4.

Table 4: Attack Success Rate for Each Attack Algorithms

| Dataset | $\ell_2$ | | | $\ell_{\inf}$ | | |
|---|---|---|---|---|---|---|
| | BB | CW | DF | PGD | BIM | FGM |
| MNIST | 1.000 | 1.000 | 0.997 | 0.983 | 0.953 | 0.773 |
| CIFAR10 | 0.978 | 0.981 | 0.908 | 0.943 | 0.967 | 0.811 |

# C   EXPERIMENTAL RESULTS FOR MNIST BENCHMARK

Table 5: Results for MNIST on L2

| Attack | Baselines | TNR@99 | | | | TNR@90 | | | | AUROC | | | | AUPR | | | | Acc | | | |
|---|---|---|---|---|---|---|---|---|---|---|---|---|---|---|---|---|---|---|---|---|---|
| | | BB | CW | DF | Ave | BB | CW | DF | Ave | BB | CW | DF | Ave | BB | CW | DF | Ave | BB | CW | DF | Ave |
| BB | Vinalla | 0.00 | 0.00 | 0.00 | 0.00 | 0.00 | 0.00 | 0.00 | 0.00 | 1.00 | 1.00 | 1.00 | 1.00 | 0.31 | 0.31 | 0.31 | 0.31 | 0.50 | 0.50 | 0.50 | 0.50 |
| | MC | 0.00 | 0.00 | 0.00 | 0.00 | 0.00 | 0.00 | 0.00 | 0.00 | 0.96 | 0.96 | 0.96 | 0.96 | 0.31 | 0.31 | 0.31 | 0.31 | 0.50 | 0.50 | 0.50 | 0.50 |
| | KD | 0.00 | 0.00 | 0.00 | 0.00 | 0.00 | 0.00 | 0.00 | 0.00 | 0.98 | 0.98 | 0.98 | 0.98 | 0.31 | 0.31 | 0.31 | 0.31 | 0.50 | 0.50 | 0.50 | 0.50 |
| | Ma | 0.18 | 0.18 | 0.18 | 0.18 | 0.43 | 0.43 | 0.43 | 0.43 | 0.69 | 0.69 | 0.69 | 0.69 | 0.61 | 0.61 | 0.61 | 0.61 | 0.67 | 0.67 | 0.67 | 0.67 |
| | tool | 1.00 | 1.00 | 1.00 | 1.00 | 1.00 | 1.00 | 1.00 | 1.00 | 1.00 | 1.00 | 1.00 | 1.00 | 1.00 | 1.00 | 1.00 | 1.00 | 1.00 | 1.00 | 1.00 | 1.00 |
| CW | Vinalla | 0.00 | 0.00 | 0.00 | 0.00 | 0.00 | 0.00 | 0.00 | 0.00 | 1.00 | 1.00 | 1.00 | 1.00 | 0.31 | 0.31 | 0.31 | 0.31 | 0.50 | 0.50 | 0.50 | 0.50 |
| | MC | 0.00 | 0.00 | 0.00 | 0.00 | 0.00 | 0.00 | 0.00 | 0.00 | 0.96 | 0.96 | 0.96 | 0.96 | 0.31 | 0.31 | 0.31 | 0.31 | 0.50 | 0.50 | 0.50 | 0.50 |
| | KD | 0.00 | 0.00 | 0.00 | 0.00 | 0.00 | 0.00 | 0.00 | 0.00 | 0.98 | 0.98 | 0.98 | 0.98 | 0.31 | 0.31 | 0.31 | 0.31 | 0.50 | 0.50 | 0.50 | 0.50 |
| | Ma | 0.21 | 0.21 | 0.21 | 0.21 | 0.43 | 0.43 | 0.43 | 0.43 | 0.70 | 0.70 | 0.70 | 0.70 | 0.62 | 0.62 | 0.62 | 0.62 | 0.67 | 0.67 | 0.67 | 0.67 |
| | tool | 1.00 | 1.00 | 1.00 | 1.00 | 1.00 | 1.00 | 1.00 | 1.00 | 1.00 | 1.00 | 1.00 | 1.00 | 1.00 | 1.00 | 1.00 | 1.00 | 1.00 | 1.00 | 1.00 | 1.00 |
| DF | Vinalla | 0.00 | 0.00 | 0.00 | 0.00 | 0.00 | 0.00 | 0.00 | 0.00 | 1.00 | 1.00 | 1.00 | 1.00 | 0.31 | 0.31 | 0.31 | 0.31 | 0.50 | 0.50 | 0.50 | 0.50 |
| | MC | 0.00 | 0.00 | 0.00 | 0.00 | 0.00 | 0.00 | 0.00 | 0.00 | 0.96 | 0.96 | 0.96 | 0.96 | 0.31 | 0.31 | 0.31 | 0.31 | 0.50 | 0.50 | 0.50 | 0.50 |
| | KD | 0.00 | 0.00 | 0.00 | 0.00 | 0.00 | 0.00 | 0.00 | 0.00 | 0.98 | 0.98 | 0.98 | 0.98 | 0.31 | 0.31 | 0.31 | 0.31 | 0.50 | 0.50 | 0.50 | 0.50 |
| | Ma | 0.19 | 0.19 | 0.19 | 0.19 | 0.42 | 0.42 | 0.42 | 0.42 | 0.68 | 0.68 | 0.68 | 0.68 | 0.60 | 0.60 | 0.60 | 0.60 | 0.66 | 0.66 | 0.66 | 0.66 |
| | tool | 0.98 | 1.00 | 1.00 | 0.99 | 1.00 | 1.00 | 1.00 | 1.00 | 1.00 | 1.00 | 1.00 | 1.00 | 1.00 | 1.00 | 1.00 | 1.00 | 0.99 | 1.00 | 1.00 | 0.99 |
| Mix | Vinalla | 0.00 | 0.00 | 0.00 | 0.00 | 0.00 | 0.00 | 0.00 | 0.00 | 1.00 | 1.00 | 1.00 | 1.00 | 0.54 | 0.54 | 0.54 | 0.54 | 0.75 | 0.75 | 0.75 | 0.75 |
| | MC | 0.00 | 0.00 | 0.00 | 0.00 | 0.00 | 0.00 | 0.00 | 0.00 | 0.96 | 0.96 | 0.96 | 0.96 | 0.55 | 0.55 | 0.55 | 0.55 | 0.75 | 0.75 | 0.75 | 0.75 |
| | KD | 0.00 | 0.00 | 0.00 | 0.00 | 0.00 | 0.00 | 0.00 | 0.00 | 0.98 | 0.98 | 0.98 | 0.98 | 0.54 | 0.54 | 0.54 | 0.54 | 0.75 | 0.75 | 0.75 | 0.75 |
| | Ma | 0.19 | 0.19 | 0.19 | 0.19 | 0.43 | 0.43 | 0.43 | 0.43 | 0.69 | 0.69 | 0.69 | 0.69 | 0.82 | 0.82 | 0.82 | 0.82 | 0.80 | 0.80 | 0.80 | 0.80 |
| | tool | 1.00 | 1.00 | 1.00 | 1.00 | 1.00 | 1.00 | 1.00 | 1.00 | 1.00 | 1.00 | 1.00 | 1.00 | 1.00 | 1.00 | 1.00 | 1.00 | 0.99 | 1.00 | 1.00 | 1.00 |

Table 6: Results for MNIST on Linf

| Attack | Baselines | TNR@99 | | | | TNR@90 | | | | AUROC | | | | AUPR | | | | Acc | | | |
|---|---|---|---|---|---|---|---|---|---|---|---|---|---|---|---|---|---|---|---|---|---|
| | | BB | CW | DF | Ave | BB | CW | DF | Ave | BB | CW | DF | Ave | BB | CW | DF | Ave | BB | CW | DF | Ave |
| BB | Vinalla | 0.02 | 0.02 | 0.02 | 0.02 | 0.14 | 0.14 | 0.14 | 0.14 | 0.58 | 0.58 | 0.58 | 0.58 | 0.57 | 0.57 | 0.57 | 0.57 | 0.58 | 0.58 | 0.58 | 0.58 |
| | MC | 0.01 | 0.01 | 0.01 | 0.01 | 0.16 | 0.16 | 0.16 | 0.16 | 0.55 | 0.55 | 0.55 | 0.55 | 0.51 | 0.51 | 0.51 | 0.51 | 0.56 | 0.56 | 0.56 | 0.56 |
| | KD | 0.00 | 0.00 | 0.00 | 0.00 | 0.07 | 0.07 | 0.07 | 0.07 | 0.64 | 0.64 | 0.64 | 0.64 | 0.40 | 0.40 | 0.40 | 0.40 | 0.50 | 0.50 | 0.50 | 0.50 |
| | Ma | 0.03 | 0.03 | 0.00 | 0.02 | 0.19 | 0.19 | 0.07 | 0.15 | 0.57 | 0.57 | 0.57 | 0.57 | 0.56 | 0.56 | 0.44 | 0.52 | 0.56 | 0.56 | 0.50 | 0.54 |
| | tool | 0.01 | 0.05 | 0.23 | 0.10 | 0.67 | 0.63 | 0.23 | 0.51 | 0.82 | 0.81 | 0.71 | 0.78 | 0.73 | 0.73 | 0.64 | 0.70 | 0.80 | 0.79 | 0.71 | 0.77 |
| CW | Vinalla | 0.03 | 0.03 | 0.03 | 0.03 | 0.16 | 0.16 | 0.16 | 0.16 | 0.59 | 0.59 | 0.59 | 0.59 | 0.58 | 0.58 | 0.58 | 0.58 | 0.58 | 0.58 | 0.58 | 0.58 |
| | MC | 0.03 | 0.03 | 0.03 | 0.03 | 0.19 | 0.19 | 0.19 | 0.19 | 0.56 | 0.56 | 0.56 | 0.56 | 0.53 | 0.53 | 0.53 | 0.53 | 0.57 | 0.57 | 0.57 | 0.57 |
| | KD | 0.00 | 0.00 | 0.00 | 0.00 | 0.07 | 0.07 | 0.07 | 0.07 | 0.64 | 0.64 | 0.64 | 0.64 | 0.39 | 0.39 | 0.39 | 0.39 | 0.50 | 0.50 | 0.50 | 0.50 |
| | Ma | 0.02 | 0.02 | 0.00 | 0.01 | 0.19 | 0.19 | 0.07 | 0.15 | 0.57 | 0.57 | 0.57 | 0.57 | 0.56 | 0.56 | 0.44 | 0.52 | 0.56 | 0.56 | 0.50 | 0.54 |
| | tool | 0.01 | 0.05 | 0.23 | 0.10 | 0.54 | 0.36 | 0.23 | 0.38 | 0.78 | 0.77 | 0.71 | 0.75 | 0.70 | 0.70 | 0.64 | 0.68 | 0.76 | 0.77 | 0.71 | 0.75 |
| DF | Vinalla | 0.00 | 0.00 | 0.00 | 0.00 | 0.00 | 0.00 | 0.00 | 0.00 | 0.88 | 0.88 | 0.88 | 0.88 | 0.34 | 0.34 | 0.34 | 0.34 | 0.50 | 0.50 | 0.50 | 0.50 |
| | MC | 0.00 | 0.00 | 0.00 | 0.00 | 0.00 | 0.00 | 0.00 | 0.00 | 0.83 | 0.83 | 0.83 | 0.83 | 0.35 | 0.35 | 0.35 | 0.35 | 0.50 | 0.50 | 0.50 | 0.50 |
| | KD | 0.00 | 0.00 | 0.00 | 0.00 | 0.00 | 0.00 | 0.00 | 0.00 | 0.92 | 0.92 | 0.92 | 0.92 | 0.32 | 0.32 | 0.32 | 0.32 | 0.50 | 0.50 | 0.50 | 0.50 |
| | Ma | 0.01 | 0.01 | 0.06 | 0.02 | 0.25 | 0.25 | 0.26 | 0.25 | 0.51 | 0.51 | 0.51 | 0.51 | 0.45 | 0.45 | 0.47 | 0.46 | 0.58 | 0.58 | 0.59 | 0.58 |
| | tool | 0.01 | 0.05 | 0.23 | 0.10 | 0.77 | 0.78 | 0.72 | 0.75 | 0.91 | 0.91 | 0.90 | 0.91 | 0.92 | 0.92 | 0.91 | 0.92 | 0.87 | 0.88 | 0.86 | 0.87 |
| Mix | Vinalla | 0.00 | 0.00 | 0.00 | 0.00 | 0.00 | 0.00 | 0.00 | 0.00 | 0.57 | 0.57 | 0.57 | 0.57 | 0.74 | 0.74 | 0.74 | 0.74 | 0.75 | 0.75 | 0.75 | 0.75 |
| | MC | 0.00 | 0.00 | 0.00 | 0.00 | 0.02 | 0.02 | 0.02 | 0.02 | 0.57 | 0.57 | 0.57 | 0.57 | 0.72 | 0.72 | 0.72 | 0.72 | 0.75 | 0.75 | 0.75 | 0.75 |
| | KD | 0.00 | 0.00 | 0.00 | 0.00 | 0.01 | 0.01 | 0.01 | 0.01 | 0.73 | 0.73 | 0.73 | 0.73 | 0.62 | 0.62 | 0.62 | 0.62 | 0.75 | 0.75 | 0.75 | 0.75 |
| | Ma | 0.02 | 0.02 | 0.00 | 0.02 | 0.22 | 0.22 | 0.11 | 0.18 | 0.54 | 0.54 | 0.54 | 0.54 | 0.76 | 0.76 | 0.71 | 0.74 | 0.75 | 0.75 | 0.75 | 0.75 |
| | tool | 0.01 | 0.05 | 0.23 | 0.10 | 0.62 | 0.63 | 0.23 | 0.49 | 0.84 | 0.83 | 0.77 | 0.81 | 0.92 | 0.92 | 0.90 | 0.92 | 0.85 | 0.84 | 0.78 | 0.82 |

