# OpenReview forum: "AttackDist: Characterizing Zero-day Adversarial Samples by Counter Attack"
_ICLR.cc/2021/Conference — Reject_

### Official Review · AnonReviewer2 · 2020-10-23
**Insufficient experiments and unreasonable assumption**

**Rating:** 3
**Confidence:** 5

**Review:**

This paper proposes to use a counterattack strategy to attack an input x, and calculate the distance between x and the crafted example x' as the detection metric. There are two main concerns about this paper:

1. The authors claim that their detection method is attack-agnostic, but their motivation in Fig 1 highly depends on the assumption of the attacking mechanism. There is no guarantee on the effectiveness of AttackDist when the attacks do not follow assumed patterns.

2. In experiments, there are only oblivious attacks, where the adversaries do not know the mechanism of AttackDist. For a defense method, it is necessary to carefully design a convinced adaptive attack and demonstrate the effectiveness of the defense under the adaptive attack.

Minors:
In the introduction section, the authors claim that "Existing adversarial defense techniques could be classified into two main categories: adversarial training and detection". Actually, there are many other types of defenses including input processing, robust architecture, random smoothing, certified defenses, etc. Besides, existing adversarial training methods like FastAT can easily scale to ImageNet, running for several hours on a single GPU. The authors should be more updated on these related progresses.

---

### Official Review · AnonReviewer3 · 2020-10-28
**Review for Paper2915**

**Rating:** 3
**Confidence:** 5

**Review:**

This paper proposes a method to detect adversarial examples. The detection
scheme is based on the observations that typical adversarial attacks generate
adversarial examples on the decision boundary, so if we use a "counter attack"
on the adversarial example, it will be easy to change its label.

A main weakness of this paper is that the proposed approach does not include an
adaptive attack for evaluation.  If the proposed detection scheme is known to
the attacker, the attacker can still generate visually indistinguishable
adversarial examples that the detector fails to detect. This can usually be
done by adding the detection objective to the loss function for attack.  Many
heuristic adversarial example detections and defense methods have been broken
by stronger and adaptive attacks [1,2], and the use of adaptive attacks is
crucial [3].

Additionally, although the paper claims to detect zero-day, or unknown attacks,
in evaluation the selection of attacks are quite limited. For example, it only
includes gradient based attacks, but not decision based attacks or evolutional
adversarial attacks.

The paper attempts to make several theoretical justifications, but these
theorems are too simple (e.g., based on direct application of triangle
inequality) and do not significantly improve the contribution of this paper.

As a conclusion, I cannot support the acceptance of this paper because the
novelty of the proposed method is limited and evaluation is insufficient.


[1] Athalye, Anish, Nicholas Carlini, and David Wagner. "Obfuscated gradients give a false sense of security: Circumventing defenses to adversarial examples." arXiv preprint arXiv:1802.00420 (2018).

[2] Carlini, Nicholas, and David Wagner. "Adversarial examples are not easily detected: Bypassing ten detection methods." Proceedings of the 10th ACM Workshop on Artificial Intelligence and Security. 2017.

[3] Carlini, Nicholas, et al. "On evaluating adversarial robustness." arXiv preprint arXiv:1902.06705 (2019).

[4] Brendel, Wieland, Jonas Rauber, and Matthias Bethge. "Decision-based adversarial attacks: Reliable attacks against black-box machine learning models." arXiv preprint arXiv:1712.04248 (2017).

[5] Cheng, Minhao, et al. "Query-efficient hard-label black-box attack: An optimization-based approach." arXiv preprint arXiv:1807.04457 (2018).

[6] Alzantot, Moustafa, et al. "Genattack: Practical black-box attacks with gradient-free optimization." Proceedings of the Genetic and Evolutionary Computation Conference. 2019.

---

### Official Review · AnonReviewer4 · 2020-10-30
**NA**

**Rating:** 5
**Confidence:** 3

**Review:**

Summary: this paper is about adversarial detection based on counter-attack. The main intuition is that the adversarial sample lies closer to the decision boundary and hence if we do counter-attack to the data, the perturbation is expected to be much smaller than the clean data.

Although the idea sounds interesting, I have a few questions to be answered:
1. in eq(1) you already defined $\Delta$ as the minimum distortion, how come it is larger than $\delta^*$ in eq(3)?
2. you assumed the $\| y-x \|_p=D(x)\sim N(\mu, \sigma^2)$, have you verified this empirically?
3. since $D(x)$ is obtained with optimal perturbation, i.e. $\|\delta^*\|$, how do you obtain the $\mu$ and $\sigma$?
4. from theorem (3) it seems that this detection method works for advanced attack, which requires $r_2 < r_1$, what if the attacker is a bad algorithm causing adversarial samples far from the boundary?

---

### Official Review · AnonReviewer1 · 2020-11-01
**Confusion in the proof of lemma 1**

**Rating:** 5
**Confidence:** 4

**Review:**

Summary:

The authors propose a novel approach to detect adversarial examples by measuring the perturbation norm after a counterattack. They theoretically provide a certified detecting performance and empirically show that AttackDist can characterize zero-day adversarial samples.
Strength:
1. The paper is well-organized and easy to understand.
2. Zero-day attacks are challenging but more practical than other adversarial samples detection problems. AttackDist can perform significantly better than the baselines to distinguish zero-day adv examples.
3. The authors show experiment results across different attacks and adversarial distances; AttackDist is consistently better than baselines.


Questions:
The idea of this paper is interesting and i would like to raise my rating if the authors can clarify my questions.

1. The proof of lemma 1 is the key to developing AttackDist. However, I am confused about 'Due to the continuous of f, …'. My question is how you make sure there always exists the point P between x and x*, such that g(P)=0; what if the deep neural network output is bounded.
2. Why not evaluate the BlackBox attacking approaches like ZOO and NATTACK[1,2]? Especially, NATTACK learns adversarial examples' distributions, and AttackDist is based on adversarial samples' distribution.
3. I am curious about whether AttackDist still works for large margin-based classifiers; what if the deep neural network is optimized with large margin loss function.



[1] Chen, Pin-Yu, et al. "Zoo: Zeroth order optimization based black-box attacks to deep neural networks without training substitute models." Proceedings of the 10th ACM Workshop on Artificial Intelligence and Security. 2017.
[2] Li, Yandong, et al. "Nattack: Learning the distributions of adversarial examples for an improved black-box attack on deep neural networks." arXiv preprint arXiv:1905.00441 (2019).

---

### Public Comment · ~Nicholas_Carlini1 · 2020-11-12
**Is it fundamental that adversarial examples are close to the decision boundary?**

All of the attacks studied are attacks explicitly designed to find the decision boundary of a neural network. There are other attacks without this objective--for example PGD from Madry et al. 2017 as mentioned in this paper. Does the proposed technique work at defending against these attacks, too? Table 1,2,5,6 all only use the boundary-finding attacks. Did I miss these results somewhere?

---

> ### Author Response · Authors · 2020-11-24
> **Response to Nicholas Carlini**
>
> Thank you for pointing out the mistake in our current version. We make a mistake in plotting Table 2 and Table 6. The results in these tables are for detecting $l_{\inf}$ attacks. As we state in the experimental setup, the $l_{\inf}$ attacks considered are PGD, BIM, and FGSM rather than BB, CW, DF. (We use the same table format from Table 1 and forget to replace the attack algorithm name) The considered $l_{\inf}$  attacks are not designed to find the decision boundary of a neural network.
> Actually, for these attacks, our approach detection performance is not as good as detecting boundary-based attacks. We discussed why the performance drops in section 4.3.2.

---

### Decision · Program_Chairs · 2021-01-07
**Final Decision**

**Decision:**

Reject

**Comment:**

Reviewers liked the concept of the zero-day attack and yet raised different concerns about the other parts of the paper. In general, Reviewers wanted to see more thorough experimental evaluations (e.g., against blackbox attack and adaptive attack) and improved clarity of the theoretical analyses. AC encourages authors to incorporate Reviewers' comments when preparing the paper for elsewhere.